# Age-Specific Mortality Forecasting in Kazakhstan: Alternative Approaches to the Lee–Carter Model

**DOI:** 10.3390/ijerph22030346

**Published:** 2025-02-26

**Authors:** Berik Koichubekov, Bauyrzhan Omarkulov, Meruyert Mukhanova, Rimma Zakirova

**Affiliations:** 1Department of Informatics and Biostatistics, Karaganda Medical University, Gogol St. 40, Karaganda 100008, Kazakhstan; zakirovar@qmu.kz; 2Department of Family Medicine, Karaganda Medical University, Gogol St. 40, Karaganda 100008, Kazakhstan; omarkulov@qmu.kz (B.O.); muhanovam@qmu.kz (M.M.)

**Keywords:** public health, mortality, mortality rate modeling, time series forecasting

## Abstract

Age-specific mortality forecasting in Kazakhstan plays a crucial role in public health planning and healthcare management. By predicting mortality rates across different age groups, policymakers, healthcare providers, and researchers can make informed decisions that improve health outcomes and allocate resources more effectively. We analyzed Kazakhstan’s annual mortality data from 1991 to 2023. The Lee–Carter model and its extensions were used to predict mortality. But they did not give satisfactory results for predicting mortality. Including external socio-economic factors in the model did not improve the forecasting accuracy. The accuracy of the forecast increased with a separate analysis of the subpopulations of children and adults. This was because, since 1991 in the children subpopulation there has been a pronounced linear downward trend, while in the adult subpopulation the global trend in mortality dynamics is nonlinear. As a result, it is possible to make forecasts for 7 years with a high degree of accuracy (error < 10%) and forecast for the 8th, 9th, and 10th years with a “good” degree of accuracy (error 10–20%). In 2024–2033, a further mortality decline is expected in most age groups. Only in groups over 80 years old is a slight increase in mortality predicted in the coming year, but then a downward trend will be observed again.

## 1. Introduction

Age-specific mortality forecasting plays a crucial role in public health planning and healthcare management. By predicting mortality rates across different age groups, policymakers, healthcare providers, and researchers can make informed decisions that improve health outcomes and allocate resources more effectively. The significance of forecasting age-specific mortality rates lies in several key areas:

Mortality prediction allows healthcare systems to anticipate the future demand for medical services, facilities, and healthcare professionals. Accurate forecasts enable health planners to allocate resources more efficiently, ensuring that adequate infrastructure and healthcare services are available to meet the needs of different age groups [1]. In addition, it helps public health authorities design targeted intervention programs. These targeted strategies can help reduce the burden of disease and lower mortality rates in vulnerable age groups [2]. Age-specific mortality forecasting is also essential for actuarial science, particularly in the insurance and pension sectors [3].

Knowledge in this area provides valuable insights into the potential impact of emerging health threats, such as pandemics, epidemics, or environmental changes. This information helps epidemiologists and public health researchers model the spread and impact of infectious diseases and assess the effectiveness of public health interventions [4]. It also helps to reveal disparities in health outcomes across different demographic and socio-economic groups. By focusing on vulnerable populations, healthcare systems can work towards achieving equity in health outcomes [5].

Another area where mortality data are needed is strategic planning at the national and regional levels. Governments can use this information to plan for future healthcare infrastructure needs, budget allocations, and workforce training. Accurate mortality predictions help align health system capabilities with future demographic trends, ultimately enhancing the resilience and sustainability of healthcare systems [6,7].

All these provisions are also relevant for Kazakhstan. However, in the literature available to us, we found only a few works devoted to the issues of mortality in Kazakhstan. Sapin A.M., using the Lee–Carter model, estimated the impact of mortality on the pension system of Kazakhstan. Unfortunately, the author does not provide data on fitting accuracy and forecasting accuracy, so it is impossible to assess how valid the model used is [8]. Other studies focus on mortality analysis from various factors and do not consider any prediction models.

For many years, the Lee–Carter mortality forecasting methodology [9] has been the reference method for extrapolative mortality forecasting. The distinctive advantages of the Lee–Carter model are that it incorporates a simple stochastic model with one time-varying parameter; it works relatively well when past trends are linear; and it is able to forecast changes in the age structure of mortality [10].

The Lee–Carter mortality model is widely used to predict mortality and construct life tables. However, despite its popularity, it has a number of shortcomings that are discussed in the scientific literature. The main ones are as follows:-Linear dependence of the mortality trend. The Lee–Carter model assumes that mortality changes linearly over time through the trend parameter kt. However, as shown in a number of studies, the actual process of mortality change may be nonlinear. This makes the model ineffective in predicting long-term trends, especially during periods of significant changes in mortality (epidemics, wars, technological breakthroughs in medicine, etc.) [11];-Limited flexibility for different age groups. The Lee–Carter model uses the fixed age parameters αx and βx, which does not always accurately reflect heterogeneous changes in mortality in different age groups. This may lead to inaccurate forecasts, especially for younger or older age groups, whose mortality may change more dynamically or unpredictably [12];-Insufficient attention to temporal correlation. The Lee–Carter model assumes that the temporal dynamics of the trend kt is modeled using a random walk, which assumes the absence of temporal correlation or other structural changes in the time series. In reality, the mortality trend may exhibit more complex patterns, such as autocorrelation or cyclicity [13];-Problems with long-term forecasting. Linear extrapolation of the mortality trend may lead to over- or underpredictions over long time horizons. This is because the model does not account for possible changes in the rate of mortality decline or unexpected increases in mortality [14];-Instability with small data. The Lee–Carter model requires a large amount of data to accurately calibrate its parameters, which can be a problem for small or sparse populations. In small samples, the model may produce unstable or inaccurate results [15];

For this reason, many extensions of the Lee-Carter model have been developed. The functional approach to data proposed by Hyndman and Ullah was developed as an alternative to the Lee–Carter model. This approach includes higher-order principal components and nonparametric smoothing and provides a more realistic future age structure for mortality [16].

Unlike parametric models, which require explicit assumptions about the underlying structure of the data, the Hyndman and Ullah model can accommodate a variety of patterns in the data, such as different shapes and trends in mortality rates across age groups and over time. This flexibility makes it suitable for a wide range of populations and datasets with different mortality dynamics.

Using Functional Principal Component Analysis (FPCA), the model reduces the complexity of multivariate mortality data to a few principal components that capture the most significant changes. This approach simplifies the modeling process and reduces noise, resulting in more accurate and interpretable forecasts [17]. The model provides more robust forecasts than models that only consider linear trends or are restricted to certain parametric shapes [4]. The model is relatively simple to implement, and the resulting principal components can be interpreted in terms of changes in the level, slope, and curvature of mortality rates over time. This interpretability is a key benefit for population researchers and policymakers who require clear and actionable insights from mortality forecasts.

For forecasting based on FPCA, various methods are used, among which are autoregressive models (ARIMAs), moving average or exponential smoothing (ETS) models, regression models, machine learning, etc. The use of a particular method depends on the mortality structure, transition processes in demographic and epidemiological processes, the age structure of the population, the state of the healthcare system and the socio-economic conditions of the population [11,12,18].

In their work, Hyndman and Ullah analyzed fertility data for age groups (15–19, 20–24, 25–29, 30–34, 35–39, 40–44, 45–49). The authors noted that “the approach is a natural extension of methods for mortality and fertility forecasting that have evolved over the last two decades” but did not specify which age groups could be modeled when studying mortality. However, already in their work [19], the author of the method includes individuals aged 0 to 100 years in the model. A number of authors, using the Hyndman and Ullah method, also included younger, older, and elderly groups in the model [20,21,22,23].

The aim of this study was to summarize the results of using the Lee–Carter model and its extensions, taking into account socio-economic factors, to predict mortality in Kazakhstan for ten years up to 2033. For this purpose, at the first stage we analyzed the main trends in mortality among different age groups since the formation of the Republic of Kazakhstan in 1991. Then we used the Lee–Carter model and its modification proposed by Hyndman and Ullah. The forecast was carried out with and without considering the influence of external factors. The best forecasting results were obtained by us with a separate analysis of the subpopulation of children and adults, in which different time trends were observed in previous years. When forecasting 10 years ahead, the error was no more than 20% for all age groups.

## 2. Materials and Methods

### 2.1. Data

We considered Kazakhstan age-specific mortality rates, observed annually from 1991 (since the creation of an independent state) to 2023, obtained from the website stat.gov.kz. Actual mortality rates are estimated as the ratio of number of deaths to the midyear population size for 5-year (0, 1–4, 5–9, … 80–84, 85+) age groups.

Figure 1 shows the dynamics of the total mortality rate in this period. We selected years 1991–2013 as the training timeframe (fitting period) and years 2014–2023 as the testing timeframe.

During the COVID-19 pandemic, mortality increased in Kazakhstan in 2020 and 2021, although to a much smaller extent than in other countries. Considering that these outliers are not typical of mortality dynamics in recent years (since 2003, the Republic has been experiencing a stable quasi-linear downward trend), we leveled out these outliers using the interpolation method. In the Appendix A, the actual data are highlighted in red, and the smoothed data are in blue.

### 2.2. Lee–Carter Model (LC)

The model’s basic premise is that there is a linear relationship between the logarithm of age-specific mortality rates and two explanatory factors: the initial age interval *x* and time *t*. The equation that describes this fact is as follows:(1)$$ln\left(m_{x,t}\right)=a_{x}+b_{x}k_{t}+\varepsilon_{x,t}$$
where *m_x_*_,*t*_ is the mortality rate for age *x* at time *t*, *a_x_* is average mortality at age *x* over time, *b_x_* is the deviation in mortality due to changes in the *k_t_* index, *k_t_* is the time-varying coefficients reflecting the temporal variation in mortality, and *ε_x_*_,*t*_ is the error term.

The parameters of the Lee–Carter model were estimated using the singular value decomposition (SVD) method to a matrix of centered log mortality rates [9]. Then, we utilized the ARIMA method to predict the time index *k_t_*.

### 2.3. Hyndman–Ullah Model (HU)

This model can be expressed as(2)$$\log\left(m_{x,t}\right)=\mu \left(x\right)+\sum_{k=1}^{K}\beta_{k}(x)\kappa_{k,t}+\varepsilon (x,t)$$
where *m_x,t_* is the mortality rate for age *x* at time *t*, *μ*(*x*) is the mean function of mortality over time, *β_k_*(*x*) are the functional principal component (FPC) loadings capturing age-specific patterns of mortality change, *κ_k_*_,*t*_ are the time-varying coefficients (scores) associated with each FPC reflecting the temporal variation in mortality, and *ε*(*x*,*t*) is the error term.

Following the application of the Hyndman–Ullah model, we performed Functional Principal Component Analysis (FPCA) to further reduce the dimensionality of the mortality data and retain the most significant patterns.

The prerequisites for applying FPCA are as follows:Nature of the data: The data are presented as functional dependencies (age and time profiles), which makes FPCA an appropriate method;Smoothing: The data are pre-smoothed with orthogonal functions to present them in functional form;The suitability of the method is confirmed if FPCA reveals that a significant proportion of the variance in the data is explained by a few principal components;Interpretability of the principal components: The principal components have a clear interpretation related to demographic processes;Comparison with peers: FPCA is similar to PCA and EFA but is designed for functional data, making it a better choice in this case;Fitting results: The suitability of the method is also confirmed by the high accuracy of the fit.

We estimated the model using different basis functions:-Automatic extraction orthonormal basis functions directly from the data through FPCA. This ensures that the basis functions adapt to the underlying structure of the mortality data (package *ftsa* in R4.4.2);-B-splines;-Fourier basis;-Cubic splines.

The FPCA-transformed components (scores *κ_k_*_,*t*_) reflect the evolution in mortality trends over time and provide a compact representation of the original high-dimensional data. Several methods were used to forecast the principal component scores:-ARIMA [24] with an automatic stationarity check and selection of the best parameters (stationarity testing results are given in Appendix A);-Linear regression (LR);-Linear regression with Added Stochastic Oscillations (LR + A) is a forecasting approach that combines the deterministic modeling capabilities of linear regression with the flexibility of stochastic components to account for random fluctuations or oscillatory behaviors in the data. In our study, the stochastic component is described using ARIMA [25];-The Generalized Additive Model (GAM) is an extension of linear and generalized linear models that accommodates nonlinear relationships between predictors and the response variable. It constructs the model by additively combining smoothing functions for the predictors, making it a powerful tool for modeling complex dependencies [26];-The ETS model, also known as the Exponential Smoothing State Space Model, is a time series forecasting approach that captures three key components: error, trend, and seasonality. The model’s formulation considers different combinations of these components, which can be either additive or multiplicative [27].

### 2.4. Hyndman–Ullah Method with Regression of Scores on Exogenous Factors

To account for the influence of exogenous factors on mortality, we applied a Two-Step Functional Principal Component Regression (FPCR) approach [28,29]. In the first step, we performed the FPCA to decompose mortality rates into principal components. In the second step, instead of extrapolating scores only from their internal dynamics, we used information about exogenous factors to make a more accurate forecast. For each score, a regression was performed on external factors:(3)$$\kappa_{k,t}=\alpha +\gamma_{1}z_{1}+\gamma_{2}z_{2}+\gamma_{3}z_{3}+\ldots +\gamma_{n}z_{n}+\varepsilon_{t}$$

α—is intercept, γ_1_, γ_2_, γ_3_, …, γ_n_ are regression coefficients, *z*_1_, *z*_2_, *z*_3_, …, *z_n_* are exogenous factors, and *ε_t_* is error.

We selected population size (Popul), density of medical doctors (Doctors), and gross domestic product per capita (GDP) as exogenous factors.

In our analysis, we employed the bootstrap sampling method to artificially augment the dataset and assess the robustness of our regression models. Bootstrap is a statistical technique that involves sampling with replacements from the original dataset to create new datasets of the same or a different size. This method is particularly useful when the original sample size is small, and it is not feasible to obtain additional data [30].

### 2.5. Model Validity

To validate the model in each timeframe, we compared the model output to real data and calculated accuracy metrics:

Absolute Percentage Error (APE)(4)$$APE=\frac{\left|(m_{xt}-\hat{m_{xt}})\right|}{m_{xt}}\times 100\%$$
where $m_{xt}$—real mortality rate for age *x* at time *t*, and ${\hat{m}}_{xt}$—forecasted mortality rate for age *x* at time *t*.

Mean Absolute Percentage Error (MAPE)(5)$$MAPE=\frac{1}{n}\times \sum \frac{\left|(m_{xt}-\hat{m_{xt}})\right|}{m_{xt}}\times 100\%$$

## 3. Results

### 3.1. Analysis of Mortality Rates in Kazakhstan

Figure 2 shows the logarithmic mortality rates for Kazakhstan from 1991 to 2023. In the first years of independence, there was an increase in mortality, which we associate with the difficult socio-economic situation that arose after the collapse of the Soviet Union. Then, there was a period of stabilization, and in the early 2000s, the government took intensive measures to improve the well-being of the people and develop the healthcare system. All this has led to the fact that in recent years there has been a decrease in mortality among all age groups.

If we analyze mortality rates by age, we can see a tendency to increase after 15 years. The rate of increase is relatively high for age groups from 15 to 29 years but slows down after 30 years. Mortality reaches its peak in the oldest age group, over 85 years.

### 3.2. Fitting Whole Population Mortality Data into the Lee–Carter Model

Figure 3 presents the calculated values of the LC model parameters of the years 1991–2023. The coefficient a_x_ is calculated as the average value for all age groups for all years. Based on this parameter, it can be concluded that high mortality is observed among newborns, then it reaches a minimum in the age range of 10–14 years and increases in subsequent age groups.

The coefficient *b_x_* obtained from the SVD, on the one hand, estimates the sensitivity of mortality for each age group to change in the overall time trend. When analyzing the graph, it becomes apparent that certain age groups show higher changes in mortality rates, as indicated by their respective *b_x_* values. In particular, the rate of change is higher for the 1–4 age group, followed by a slight decrease at 4–9. It then peaks in the 20–24 age group and starts to decline again. The next maxima are observed at 50–54 and after 85 years. This comprehensive visualization allows us to understand the dynamic patterns of change in mortality rates in different age groups over the observed period.

On the other hand, this coefficient reflects the extent to which the time trend *k_t_* affects mortality for a particular age group. If *b_x_* is negative, then a decrease in *k_t_* leads to an increase in mortality for a given age group, while an increase in *k_t_* leads to a decrease. Figure 3 shows that *b_x_* takes negative values in almost all age groups. The product of these two coefficients was positive and increased until 1996, then began to decrease and stabilize for some time, and since 2008 it has become negative. That is, we can conclude that before 1996, Kazakhstan experienced an increase in mortality, then some stabilization, and then a clear downward trend can be traced until 2023. All this indicates the presence of a nonlinear trend in mortality dynamics. Apparently, this was the reason for the insufficient accuracy of forecasting on the testing timeframe (2014–2023), since, as already noted, one of the shortcomings of the Lee–Carter model is its linear nature. Table 1 shows the assessment of fit accuracy and the accuracy indicators of the model on the testing timeframe. In general, the fit results can be considered satisfactory (on average MAPE = 7.23%), although in infant groups the error was 14–16%. But on the testing timeframe, certain age groups error reached 42–43% (1–4, 30–34 groups). The on average for all groups MAPE was 15.8%.

### 3.3. Fitting Whole Population Mortality Data into the Hyndman and Ullah Model

In this model, more than one principal component is used. Higher-order terms of the principal component decomposition improve the LC model, because these additional components capture non-random patterns, which are not explained by the first principal component. Using Functional Principal Component Analysis (FPCA), a set of curves is decomposed into orthogonal functional principal components and their uncorrelated principal component scores.

Various orthogonal functions can be used as basis functions. The peculiarity of the *ftsa* package is that by default the model extracts orthonormal basis functions directly from the data through FPCA. Scores were forecast using the ARIMA method with automatic selection of the best parameters. After that, we independently installed B-splines, the Fourier basis, and Cubic splines as basis functions.

At the first stage, we included all 19 age groups in the analysis. The model was fitted to Kazakhstan data from 1991 to 2013. Regardless of the type of basis functions, we obtained the same result for the contribution of the first three principal components. The basis functions explain 67%, 27.3%, and 2.5% of the variation, respectively.

The parameter *µ*(*x*) in the Hyndman and Ullah model is analogous to the coefficient ax in the Lee–Carter model (note that in the case of ax (Figure 3) we are dealing with ln(mortality rate) and in the case of *µ*(*x*) with mortality rates). It shows that mortality is high among newborns, then decreases, and increases again after 5 years (Figure 4).

The first principal component *β*_1_(*x*) characterizes the contribution of each age group to the overall mortality trend. In the children’s subpopulation, this contribution decreases with age, while in the adult subpopulation, it increases. *β*_1_(*x*), together with the scores *k*_1*t*_, determine how this contribution changes in different age groups over time. These two values characterize the interaction between age and time trends. According to them, the global mortality trend is expressed in its growth until approximately 1998, then a slow decline until 2006 and an acceleration in this trend thereafter.

The second component *β*_2_(*x*) explains local changes in mortality in a separate age group associated with the influencing factors characteristic of it. Such changes are observed in the 0–4, 20–24, and 50–54 age groups. According to the scores’ *k*_2*t*_ values, before 2006 these were the factors that increased mortality, but later mortality was influenced by factors that reduced it.

The contribution of the third component is only 2.5%, and its interpretation is not obvious. Table 2 provides summaries of the point fitting and forecasting accuracy based on the MAPE of age group mortality rates averaged over different years in the training and testing periods. We found that regardless of the type of basis functions, the fitting results were the same. HU methods tend to perform with better accuracy than the LC methods. In the training timeframe, the MAPE values ranged from 1.39% to 7.56% for different age groups. On average, the fitting MAPE value was 2.94%.

However, the accuracy of forecasting was low on the test sample. The average value of MAPE for all age groups was 25.5%. The greatest discrepancy between the actual and forecast data was found in children’s age groups: in the (0) group MAPE = 94.4%, in the 1–4 age group MAPE = 49.1%, and in the 5–9 age group MAPE = 29.7%.

### 3.4. Influence of External Factors on Forecasting

External socio-economic factors have a major impact on mortality, among which the most important are the state of the healthcare system and the level of well-being of the population. In this regard, after identifying the main components, we predicted the principal component scores for the test period, considering them as a linear regression from the predictors of population size, the density of medical doctors, and gross domestic product per capita (Table 3).

As already indicated, the first component *β*_1_(*x*), together with the *k*_1*t*_ scores, determine how this contribution changes in different age groups over time. Based on the data in Table 3, we can estimate the relationship of *k*_1*t*_ with external factors. The greatest influence is exerted by GDP; almost 79% of the variance falls on it. The regression coefficient of this factor has a negative value, and, accordingly, this indicates the opposite effect of this factor on mortality. That is, an increase in GDP leads to a decrease in mortality. The next most important is the density of doctors (20.83% of the variance). The influence of population size is statistically insignificant.

The second principal component characterizes individual age group mortality’s impact on the global trend. Over time, these impacts may change, which is reflected using the score *k*_2*t*_. External predictors’ degree of influence on this coefficient can be assessed using the regression analysis data presented in Table 3.

According to these data, the greatest impact is exerted by population size (54.83% of variance) and GDP (44.97% of variance). The factor of provision of medical personnel does not have a significant impact. After obtaining the regression equations on the training timeframe, *k*_1*t*_ and *k*_2*t*_ were forecasted for 10 years (2014–2023) using the ARIMA method without external factors and the linear regression method with external factors. The results are presented in Figure 5. The dynamics of the coefficients calculated for the entire analyzed period 1991–2023 are also shown.

#### 3.4.1. Forecasting First Component Score

The gray line in Figure 5A, reflecting the dynamics of first component scores over the entire observation period, demonstrates a consistent decrease in this value on the 2014–2023 timeframe. The regression analysis showed that the contribution of GDP to the total variance (i.e., its impact on the first component) is the largest among all the factors. However, if on the training timeframe GDP demonstrated almost linear growth, then on the testing timeframe the dynamics changed to nonlinear (Figure 6A). This affected the forecasting of first component scores. The green dotted line in Figure 5A shows the nonlinear nature of the forecast.

In the same Figure 5A, the blue dotted line is the result of forecasting using the ARIMA method (excluding external factors). In this case, the forecast data are closer to the actual data.

#### 3.4.2. Forecasting the Second Component Score

The dynamics of the second component score on the training timeframe show local mortality changes’ influence on overall mortality. At first, it weakens (the score value approaches zero), then increases in the period from 1996 to 2007 (the score value becomes positive), and then decreases again starting in 2007. The second component is more influenced by the population size, which decreased until the early 2000s and has been growing in recent years (Figure 6B). This factor enters the regression equation with a minus sign, so on the test timeframe it is predicted that population growth reduces the value of the second component score.

GDP also had a strong positive relationship with the second component on the training timeframe, but apparently this relationship weakened in the test period, since GDP was nonlinear in these years. As a result, we believe that the main influencing factor was the growing population, which led to a decrease in *k*_2*t*_ (green line in Figure 5B). In reality, decline was slower (gray line), and this difference in the rate of decline became the source of the forecast error. It should be noted that in the analyzed period, the doctor density in the republic has not undergone significant changes (Figure 6C). Therefore, the influence of this factor was insignificant.

After forecasting the first and second component scores on the testing timeframe, mortality for each age group was calculated and compared with real data. According to the data in Table 4, considering external factors did not lead to an improvement in forecasting.

### 3.5. Subpopulation Analysis

We found that the dynamics of the mortality rate in children in 1991–2023 had its own characteristics. In the age groups 0, 1–4, 5–9, and 10–14, there was a clear linear downward trend, with some fluctuations in individual years (Figure 7A shows examples of changes in mortality in the child subpopulation). In adults, mortality increased in the first years, then a period of relative stabilization set in, and starting in 2007, a downward trend emerged (Figure 7B shows examples of changes in mortality in the adult subpopulation).

Based on these results, we analyzed mortality prediction models separately for the subpopulations of children and adolescents–adults.

#### 3.5.1. Fitting Adolescent–Adult Subpopulation Mortality Data into the Hyndman and Ullah Model

The adolescent–adult subpopulation included age groups 15–19 and older. In the ftsa package, the extracting orthonormal basis functions directly from the data through the FPCA and ARIMA forecasting mode was used. The accuracy of the fit on the training timeframe was quite high; the average MAPE value for all age groups was 2.29%. On the testing timeframe, the forecasting accuracy averaged about 90%, which also satisfied us (Table 5).

#### 3.5.2. Fitting Child Subpopulation Mortality Data

The children’s subpopulation included four groups, and therefore we could not use the Hyndman and Ullah model due to insufficient data. Therefore, a time series analysis model was selected for each group separately. Among them were ARIMA, ETS, GAM, LR, and LR + A. Figure 8 shows the validation results of various mathematical models. The blue line represents the actual data, the yellow line represents the best fitting result on a training timeframe, and the dotted lines represent the results of forecasting on a testing timeframe using different methods.

#### 3.5.3. Mortality Prediction Model for Group (0)

Table 6 and Figure 8 shows that in the test area, the best prediction result was obtained using the LR + A method—on average, the APE was 14.17%. However, if we look at the years, the ARIMA results are preferable, since in the first five years there is a prediction error MAPE < 10%, while according to the LR + A method, the prediction accuracy in the first five years was in the range of a MAPE from 10 to 20%.

#### 3.5.4. Mortality Prediction Model for the 1–4 Age Group

For this group, the GAM method (Table 7, Figure 8) showed the best result on the test section. The average APE value was 17.13%. A high prediction accuracy (APE < 10%) was observed in the first five years. In the following two years, the results can be considered satisfactory. In general, this method shows the possibility of making predictions for 7 years (MAPE = 8.7%).

#### 3.5.5. Mortality Forecasting Model for the 5–9 Age Group

In the 5–9 age group, the best results were shown by the linear regression and regression with ARIMA methods (Table 8, Figure 8). The average forecast error in the 10-year test section was 8.23%. We achieved a MAPE < 10% in the timeframe from 2014 to 2020. In other years, the forecast accuracy was at least 80%.

#### 3.5.6. Mortality Forecasting Model for the 10–14 Age Group

As can be seen from Table 9 and Figure 8, in the 10–14 age group, linear methods also showed a good result in predicting mortality. When projecting mortality in 2015 and over the time interval from 2018 to 2023, the APE was less than 10%. The forecast for other years had an error from 11.48% to 14.29%.

### 3.6. Mortality Forecasting Up to 2033

Based on the selected models, mortality in Kazakhstan was forecasted until 2033.

As Figure 9 shows, we expect further declines in mortality in most age groups. Only in groups over 80 years old is a slight increase in mortality predicted in the coming year, but then a downward trend will be observed again. The forecast data are given in the Appendix A.

## 4. Discussion

The Lee–Carter model and its modifications are particularly useful when global trends affecting all age groups need to be considered, the consistency of predictions across ages is important, and an interpretable and efficient model is required.

This paper presents an applied approach to forecasting age-specific mortality series using principal component analysis and comparing the predicted continuous distribution rather than the observed discrete values. In general, Hyndman and Ullah discussed that this approach yields better forecasting results than the Lee and Carter approach to mortality forecasting. However, the success of a particular method often depends on the specific dynamics of the mortality rate in a given country and on the quality of the available data. We examined this method using data on annual age-specific mortality among the population of Kazakhstan. Our main goal was to forecast mortality in Kazakhstan for ten years up to 2033, since this is important both in economic and social terms.

Estimates of the parameters of both models indicate that in Kazakhstan, age-specific mortality is similar to the one observed in most of the countries in which this analysis was carried out—high mortality at birth, a decrease in children’s groups, and an increase with age in the adult subpopulation. Both models show a maximal rate of change in mortality in the 0–4, 20–24, and 50–54 age groups. Overall, mortality dynamics are nonlinear, and this may be the reason why both models yielded poor prediction results on the testing timeframe.

Over the past 15 years, Kazakhstan has seen a consistent decline in mortality. This is due to the overall growth of economic and social well-being and the implementation of various programs to improve health services to the population. Special measures have been developed and implemented in the field of motherhood and childhood. In this regard, we assumed that considering external factors in the model would improve its predictive ability. Unfortunately, we were limited in the choice of these external factors, since statistical data on the state of the economy in general, and healthcare in particular, began to be officially published only since the early 2000s. We had access to data on population, GDP, and the provision of medical specialists. However, including these factors in the model did not improve forecasting. One of the reasons is the change in GDP during periods of crisis in the economy. In 1991–2013, GDP showed confident growth, and its correlation with mortality was high, more than −0.8. Subsequently, there were periods of both a decrease and an increase in GDP, so the correlation coefficient decreased to −0.11.

Subsequently, we suggested that our forecasting failures were due to the peculiar properties of the dynamics of mortality in different subpopulations.

It should be noted that several studies in the literature suggest that the classical Lee–Carter model sometimes performs as well as or even better than its extensions or alternative models. A study [31] showed that while deep learning models exhibited certain advantages, the Lee–Carter model combined with ARIMA provided the best overall balance of forecast accuracy, computational efficiency, and interpretability. Another study evaluated different variants of the Lee–Carter model. While these extended models generally provided slightly better predictive accuracy, the improvement was often marginal. In some cases, the original Lee–Carter model performed just as well as its modified versions [32].

However, the LC model assumes that mortality changes linearly over time through the trend parameter kt. In the HU model, if the trends in age groups differ greatly (for example, one age group has a linear trend, and another a nonlinear one), FPCA may inadequately identify common trends, and some of the significant dynamics will remain in the residuals ε(x,t). Particularly, high error values were obtained in children’s groups, up to 15 years old. In this context, we analyzed the mortality curves separately in the children and adult subpopulations. We have seen that in the children subpopulation since 1991, there has been a pronounced linear downward trend, while in the adolescent–adult subpopulation the global trend in mortality dynamics is nonlinear. Based on this, it was decided to conduct the analysis separately among children and adolescents–adults. As a result, in the adolescent–adult subpopulation, the Hyndman and Ullah model showed satisfactory results both at the fitting and testing stages.

Only four groups were included in the children subpopulation. The use of small samples in the Lee–Carter and Hyndman and Ullah models is associated with a number of problems:-Small samples provide little information about the age structure of mortality, making it difficult to model complex interactions. As a result, important information about mortality in adjacent ages may be missed, especially if the mortality dynamics vary greatly between age groups;-A small sample is more sensitive to noise and the probability that random fluctuations in mortality will be interpreted by the model as significant trends increases;-The amount of information available for identifying time trends is limited, which leads to uncertainty in forecasts. Forecasts may be unstable or unrealistic;-With a small number of groups, the interpretation of components becomes difficult, since they may not have a clear biological or demographic meaning;

In this context, it was decided to conduct the analysis separately in each group using different time series methods. We assessed which methods give the smallest APE error on the test sample.

In the period from 1995 to 2013, the decline in mortality in childhood age groups occurred with high intensity; the annual rate of decline over these years averaged 5.1%. Apparently, this was due to the general improvement in the economic situation in Kazakhstan and the implementation of a number of state programs in the field of healthcare and especially in the field of maternal and child health. In subsequent years, this rate slowed down to 3.9%. The models used “could not” adequately reflect this inflection point, “trying to maintain the gained rate” for the future, which led to a mismatch between the real and model data. This mismatch increased as the forecasting periods increased. We selected the models that showed the greatest accuracy. They turned out to be ARIMA for the age group (0), GAM for the 1–4 group, and LR for groups 5–9 and 10–14. Using them, it is possible to make forecasts for 7 years with a high degree of accuracy (error <10%) and forecast for the 8th, 9th and 10th years with a “good” degree of accuracy (error 10–20%).

## 5. Conclusions

The Lee–Carter model was first used for US mortality data. Later, the use of the model for different countries showed that the basic version sometimes transfers poorly to data from other regions, especially in countries with differences in mortality levels and structure. In this regard, various modifications of it appeared. We showed possible ways of applying the model to Kazakhstan data and predicted mortality until 2033. The main conclusion is that the emerging trend towards a decrease in mortality will continue in the next decade.

Despite the importance of the results obtained, our study is not without a number of limitations. This article does not present data separately for female and male populations. A preliminary analysis showed that there are no significant gender differences in the age and time structure of mortality in Kazakhstan. In order not to overload our article with numerous tables and figures, we decided to consider this aspect in more detail in future publications.

As already mentioned above, over the past 18 years, the Republic has seen a consistent decrease in mortality in all age groups. Obviously, this is happening under the influence of controllable socio-economic factors. For future work, taking these factors into account may improve the model fit and increase prediction accuracy.

Another limitation is that our work did not perform the actuarial standardization procedure, which ensures the uniformity, accuracy, and comparability of results in the field of actuarial calculations. In subsequent works, both we and other authors need to take this aspect into account when modeling in demographic terms.

In our study, the training timeframe included 23 years starting in 1991, from the moment of the formation of the independent Republic of Kazakhstan. At the same time, the WHO website contains data on mortality in Kazakhstan starting from 1950, when the Republic was still part of the USSR. Perhaps using this data will lead to different results. This aspect can also be considered in future works.

## Figures and Tables

**Figure 1 ijerph-22-00346-f001:**
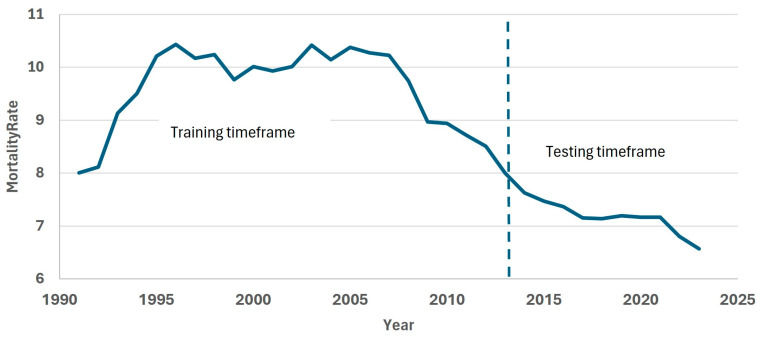
Total mortality rate from 1991 to 2023 in Kazakhstan (per 1000). The dotted line separates training and testing timeframes.

**Figure 2 ijerph-22-00346-f002:**
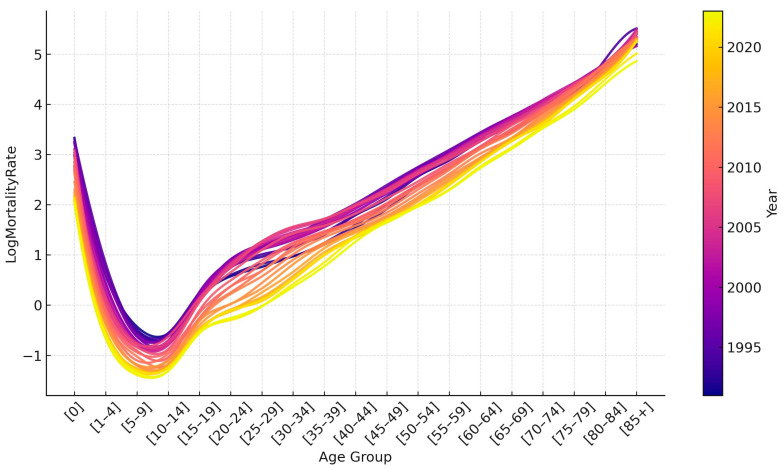
Age-specific log mortality rate in Kazakhstan from 1991 to 2023.

**Figure 3 ijerph-22-00346-f003:**
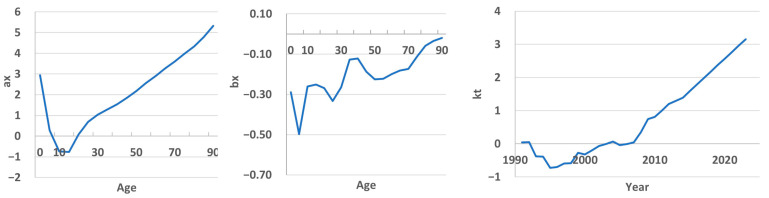
LC model parameters calculated over the years 1991–2023. *a_x_* and *b_x_* are age-dependent parameters, and *k_t_* is a specific mortality index for each year.

**Figure 4 ijerph-22-00346-f004:**
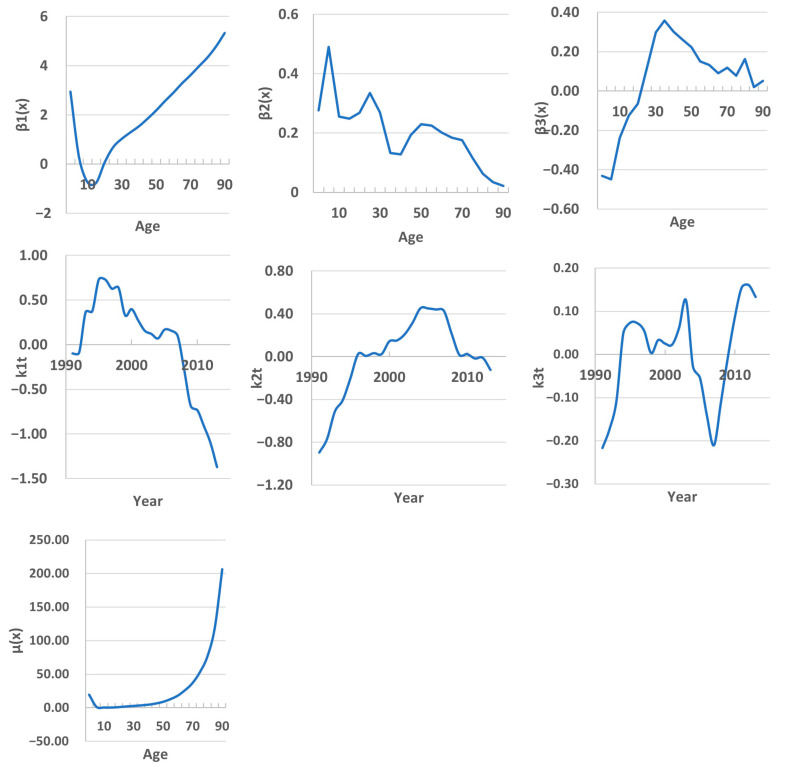
Hyndman and Ullah model parameters: *µ*(*x*) is the mean function, *β*_1,2,3_(*x*) is a set of the first 3 functional principal components; *k*_1,2,3*t*_ is a set of uncorrelated principal component scores.

**Figure 5 ijerph-22-00346-f005:**
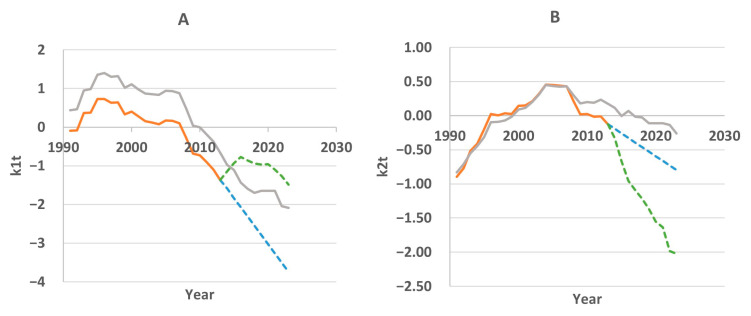
Forecasting first component score (**A**) and second component score (**B**). Orange line—score calculation on the training timeframe, blue line—forecasting without external factors, green line—forecasting with external factors, gray line—score calculation for the period 1991–2023.

**Figure 6 ijerph-22-00346-f006:**
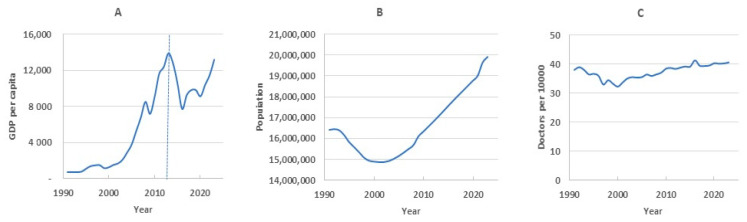
Dynamics of external factors from 1991 to 2023: GDP (**A**), population (**B**), doctor density (**C**).

**Figure 7 ijerph-22-00346-f007:**
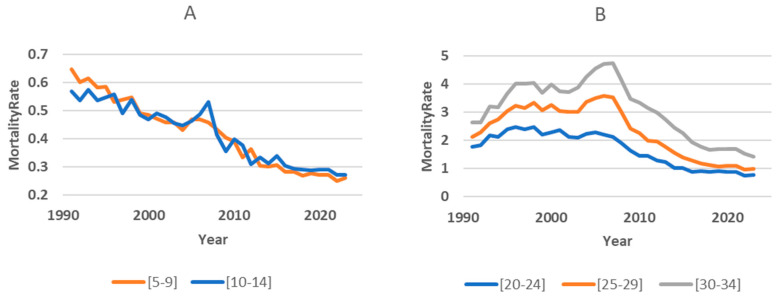
Examples of long-term mortality trends in child (**A**) and adult (**B**) subpopulations.

**Figure 8 ijerph-22-00346-f008:**
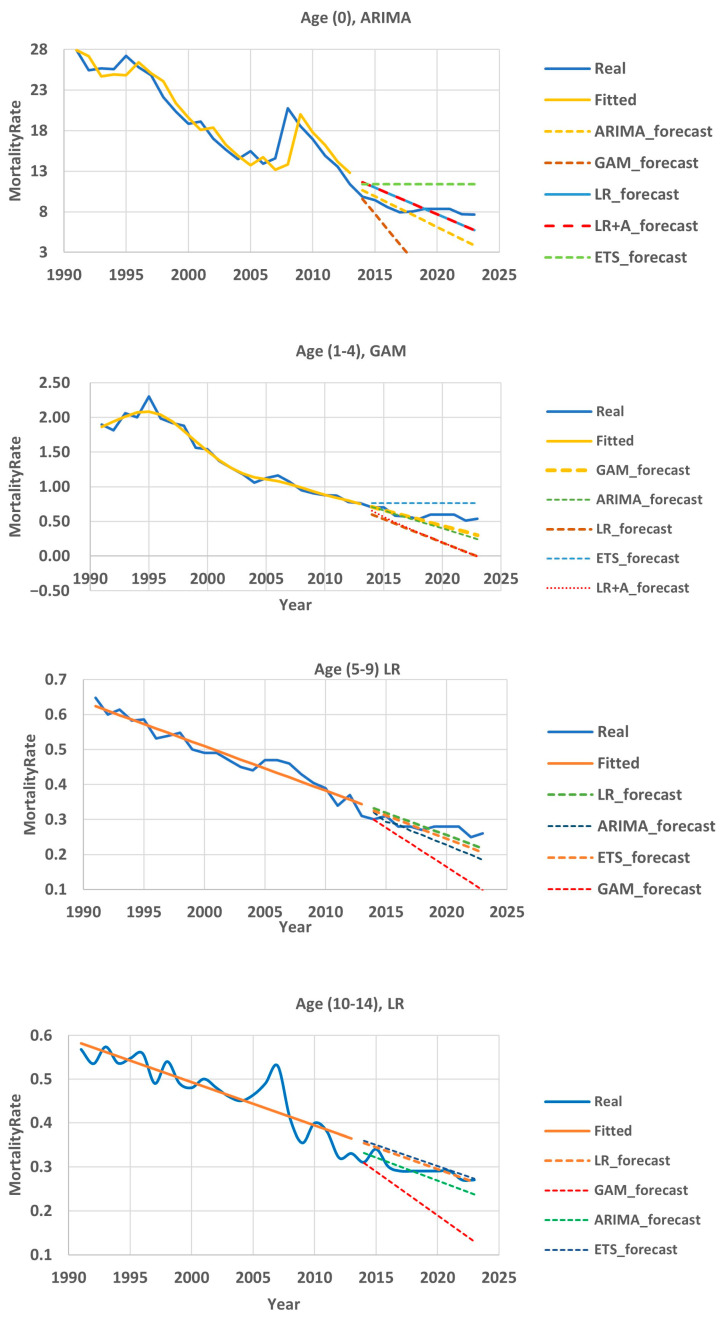
Child subpopulation mortality forecasting results. The blue line represents the actual data, the yellow line represents the best fitting result on a training timeframe, and the dotted lines represent the results of forecasting on a testing timeframe using different methods.

**Figure 9 ijerph-22-00346-f009:**
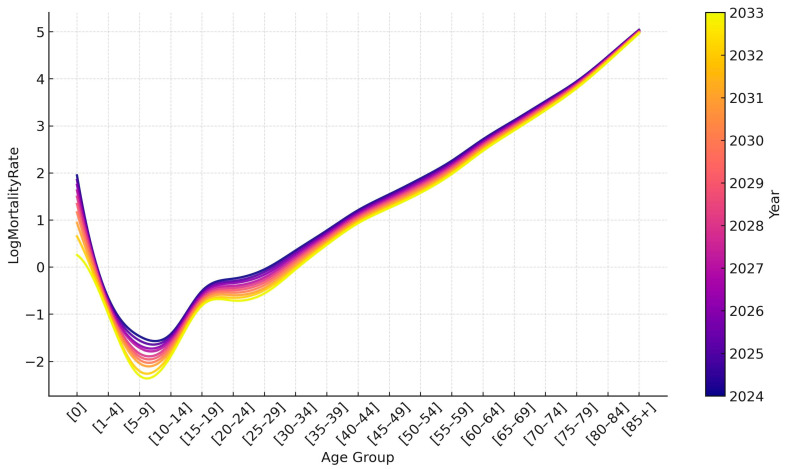
Age-specific mortality forecasting up to 2033.

**Table 1 ijerph-22-00346-t001:** Lee–Carter model accuracy.

	Training Timeframe	Testing Timeframe
Age Group	MAPE, %	MAPE, %
0	16.18	14.70
1–4	14.00	42.88
5–9	7.18	18.25
10–14	5.92	21.24
15–19	3.96	17.66
20–24	5.30	16.28
25–29	10.68	14.17
30–34	11.11	42.02
35–39	9.13	19.00
40–44	7.79	8.08
45–49	6.90	14.35
50–54	5.36	10.30
55–59	4.86	7.71
60–64	4.06	9.32
65–69	4.28	13.00
70–74	4.07	3.26
75–79	5.87	7.99
80–84	3.86	12.02
85+	6.84	8.02
Mean	7.23	15.80

**Table 2 ijerph-22-00346-t002:** Accuracy of Hyndman and Ullah model with different basis function types.

	Training Timeframe (by Default, B-Splines, Fourier Basis, Cubic Splines)	Testing Timeframe (by Default, B-Splines, Fourier Basis, Cubic Splines)
Age Group	MAPE, %	MAPE, %
0	7.56	94.4
1–4	4.09	49.1
5–9	2.52	29.7
10–14	3.81	16.0
15–19	2.77	22.1
20–24	2.57	32.1
25–29	2.41	38.3
30–34	3.01	37.9
35–39	2.68	23.5
40–44	1.63	8.0
45–49	1.69	8.1
50–54	1.81	18.3
55–59	1.39	18.7
60–64	1.68	17.5
65–69	1.49	8.3
70–74	3.01	18.2
75–79	2.98	10.1
80–84	3.10	20.9
85+	5.70	13.4
Mean	2.94	25.5

**Table 3 ijerph-22-00346-t003:** Results of regression analysis.

	*k*_1*t*_ (R^2^ = 0.84; F-Statistic *p*-Value = 6.69 × 10^−8^)			*k*_2*t*_ (R^2^ = 0.89; F-Statistic *p*-Value = 1.34 × 10^−9^)		
Factor	γ	*p*-Value	Contribution	γ	*p*-Value	Contribution
Intercept	3.707	0.008		8.256	2.48 × 10^−10^	
GDP	−9.89 × 10^−5^	5.97 × 10^−8^	78.79%	7.15 × 10^−5^	7.15 × 10^−10^	44.97%
Doctors	−0.112	0.04	20.83%	−0.011	0.7081	0.20%
Popul	4.54 × 10^−8^	0.782	0.38%	−5.23 × 10^−7^	1.09 × 10^−5^	54.83%

**Table 4 ijerph-22-00346-t004:** Accuracy of Hyndman–Ullah model with regression of scores.

	Training	Testing Without Exogenous Factors	Testing with Exogenous Factors
Age Group	MAPE, %	MAPE, %	MAPE, %
0	7.56	94.4	78.69
1–4	4.09	49.1	50.90
5–9	2.52	29.7	30.15
10–14	3.81	16.0	21.18
15–19	2.77	22.1	19.10
20–24	2.57	32.1	29.80
25–29	2.41	38.3	35.50
30–34	3.01	37.9	36.26
35–39	2.68	23.5	23.56
40–44	1.63	8.0	9.20
45–49	1.69	8.1	9.53
50–54	1.81	18.3	21.08
55–59	1.39	18.7	19.83
60–64	1.68	17.5	19.22
65–69	1.49	8.3	7.03
70–74	3.01	18.2	18.01
75–79	2.98	10.1	10.55
80–84	3.10	20.9	18.75
85+	5.70	13.4	13.87
Mean	2.94	25.5	24.85

**Table 5 ijerph-22-00346-t005:** Accuracy of adolescent–adult subpopulation mortality Hyndman and Ullah model.

	Training Timeframe	Testing Timeframe
Age Group	MAPE, %	MAPE, %
15–19	3.46	6.77
20–24	1.70	12.88
25–29	2.57	9.65
30–34	2.14	17.20
35–39	1.89	4.68
40–44	1.57	14.53
45–49	1.61	17.25
50–54	1.69	10.00
55–59	1.50	7.32
60–64	1.46	7.13
65–69	1.57	12.41
70–74	2.30	3.39
75–79	2.70	3.16
80–84	3.01	13.69
85+	5.25	7.68
Mean	2.29	9.85

**Table 6 ijerph-22-00346-t006:** Mortality prediction accuracy of different models for age group (0).

	APE, %				
Year	ARIMA	ETS	GAM	LR	LR + A
2014	8.22	15.87	2.58	18.52	12.42
2015	5.07	21.04	17.91	16.80	12.65
2016	6.35	32.60	31.63	20.27	17.31
2017	5.72	43.63	49.29	21.97	19.87
2018	4.96	41.85	72.98	12.23	10.88
2019	17.80	36.08	96.20	0.20	1.05
2020	26.78	36.08	118.33	8.08	8.63
2021	35.76	36.08	140.45	15.96	16.32
2022	39.78	48.31	168.20	17.00	17.26
2023	49.23	49.28	192.92	25.10	25.27
Mean	19.97	36.08	89.05	15.61	14.17

**Table 7 ijerph-22-00346-t007:** Mortality prediction accuracy of different models for 1–4 age group.

	APE, %				
Year	ARIMA	ETS	GAM	LR	LR + A
2014	1.19	8.57	1.23	14.53	6.36
2015	6.19	8.57	5.23	24.14	19.19
2016	4.31	31.03	6.59	20.04	16.42
2017	2.92	33.33	0.53	30.44	28.21
2018	5.35	43.40	0.40	37.88	36.43
2019	25.00	26.67	19.55	56.34	55.56
2020	33.61	26.67	27.08	67.55	67.08
2021	42.22	26.67	34.62	78.76	78.48
2022	42.16	49.02	31.94	88.20	88.00
2023	54.94	40.74	44.09	101.31	101.20
Mean	21.79	29.47	17.13	51.92	49.69

**Table 8 ijerph-22-00346-t008:** Mortality prediction accuracy of different models for 5–9 age group.

	APE, %				
Year	ARIMA	ETS	GAM	LR	LR + A
2014	6.77	8.17	0.30	10.56	10.56
2015	5.41	0.41	10.75	2.90	2.90
2016	2.15	6.43	9.19	9.39	9.39
2017	4.19	1.70	17.20	4.86	4.86
2018	5.14	0.57	22.44	4.04	4.04
2019	13.94	7.76	33.21	4.20	4.20
2020	18.78	12.49	41.22	8.74	8.74
2021	23.92	17.22	49.22	13.27	13.27
2022	20.37	12.58	52.10	7.94	7.94
2023	28.88	21.04	62.56	16.36	16.36
Mean	12.96	8.83	29.82	8.23	8.23

**Table 9 ijerph-22-00346-t009:** Mortality prediction accuracy of different models for 10–14 age group.

	APE, %				
Year	ARIMA	ETS	GAM	LR	LR + A
2014	7.03	15.85	0.28	14.29	14.29
2015	5.49	2.82	14.91	1.29	1.29
2016	3.63	13.35	10.19	11.49	11.49
2017	3.60	13.96	13.94	11.92	11.92
2018	0.00	10.66	20.79	8.50	8.50
2019	3.60	7.37	27.63	5.08	5.08
2020	7.20	4.07	34.48	1.66	1.66
2021	10.80	0.78	41.33	1.76	1.76
2022	8.07	4.71	44.34	1.85	1.85
2023	11.93	1.17	51.69	1.82	1.82
Mean	6.14	7.47	25.96	5.96	5.96

## Data Availability

This published article, along with its Appendix A, contains all the data generated or analyzed during the study. The data supporting the findings of this study are available from https://www.who.int (accessed on 1 June 2024), statgov.kz (accessed on 1 June 2024), and worldbank.org (accessed on 1 June 2024). The ftsa package is available at https://github.com/cran/ftsa (accessed on 1 June 2024).

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
