# Peer review of "Age-Specific Mortality Forecasting in Kazakhstan: Alternative Approaches to the Lee–Carter Model"

_ijerph, 2025, doi:10.3390/ijerph22030346_

Round 1
Reviewer 1 Report
Comments and Suggestions for Authors
Dear Authors,
I have attached a review of the manuscript "Age-specific Mortality Forecasting in Kazakhstan. Modeling Applying Experience".

Reviewer 2 Report
Comments and Suggestions for Authors
Review
The paper presents an interesting exploration of alternative approaches to the Lee-Carter model for age-specific mortality forecasting. However, several issues need to be addressed to improve the overall quality of the manuscript:
-
English Proficiency:
The manuscript requires significant improvement in English. It is recommended to use a professional proofreading service to enhance readability and clarity. -
Title:
The current title consists of two sentences, which is uncommon and does not clearly convey the research focus. It is suggested to revise the title as:
“Age-Specific Mortality Forecasting in Kazakhstan: Alternative Approaches to the Lee-Carter Model”. -
Abstract:
The abstract needs refinement. It should highlight the importance of age-specific mortality forecasting in Kazakhstan in the opening sentence. This will provide context and emphasize the relevance of the study. -
Data Limitation:
The observed data spans only 33 years (1991–2023), which is insufficient for forecasting 10 years into the future. In epidemiological research, it is more common to perform short-term forecasts, such as 2–3 years ahead, due to potential changes in external factors that can impact age-specific mortality rates. This limitation should be acknowledged and addressed. -
Research Argument:
The argument that previous research lacks data on fitting and forecasting accuracy, making it impossible to assess the validity of the models used, is not compelling. These metrics are relatively straightforward to calculate. A stronger and more solid justification for the research is needed. -
Model Validity and Notation:
The model's validity section requires revision. The notation yiy_i should be replaced with yx,ty_{x,t}, denoting the observed mortality for age xx at time tt, to ensure clarity and consistency. -
Figure 2:
This figure needs revision. Specifically:- The meaning of the different colors must be explained in the figure legend or caption.
- The x-axis should represent time (period) for consistency.
-
Figures:
All figures should include detailed explanations. The meaning of different colors must be explicitly clarified to avoid any confusion for the readers. -
Figure 9:
This figure should be moved to the Results section rather than the Conclusion. Additionally, the forecasted time periods should be displayed using distinct colors for better differentiation. It is recommended to use "time period" as the x-axis instead of "age-specific" for clarity.
Round 2
Reviewer 1 Report
Comments and Suggestions for Authors
Dear Authors!
I am attaching a review of the article.

Reviewer 2 Report
Comments and Suggestions for Authors
The revised manuscript shows significant improvements compared to the previous version. The authors have addressed the suggested revisions accordingly. However, the title should be adjusted as recommended. This change in the title should not pose an issue for the editor, considering that using a two-sentence title is uncommon in scientific article writing.
Author Response
Dear Reviewer, Thank you for your careful review of our article. We have agreed to change its title.